# How High Is High Enough? Assessing Financial Risk for Vertical Farms Using Imprecise Probability

Francis J. Baumont de Oliveira [1,*], Scott Ferson [1], Ronald A. D. Dyer [2], Jens M. H. Thomas [3], Paul D. Myers [3] and Nicholas G. Gray [1]

1 Institute for Risk and Uncertainty, University of Liverpool, Liverpool L69 3BX, UK; scott.ferson@liverpool.ac.uk (S.F.); nicholas.gray@liverpool.ac.uk (N.G.G.)
2 Management School, University of Sheffield, Sheffield S10 2TN, UK; ronald.dyer@sheffield.ac.uk
3 Farm Urban, Liverpool L1 0AF, UK; jens@farmurban.co.uk (J.M.H.T.); paul@farmurban.co.uk (P.D.M.)
* Correspondence: f.baumont-de-oliveir@liv.ac.uk

**Abstract:** Vertical farming (VF) is a method of indoor agricultural production, involving stacked layers of crops, utilising technologies to increase yields per unit area. However, this emerging sector has struggled with profitability and a high failure rate. Practitioners and academics call for a comprehensive economic analysis of vertical farming, but efforts have been stifled by a lack of valid and available data as existing studies are unable to address risks and uncertainty that may support risk-empowered business planning. An adaptable economic analysis is necessary that considers imprecise variables and risks. The financial risk analysis presented uses with a first-hitting-time model with probability bounds to evaluate quasi-insolvency for two unique vertical farms. The UK farm results show that capital injection, robust data collection, frequent cleaning, efficient distribution and cheaper packaging are pathways to profitability and have a safer risk profile. For the Japanese farm, diversification of revenue streams like tours or education reduce financial risk associated with yield and sales. This is the first instance of applying risk and uncertainty quantification for VF business models and it can support wider agricultural projects. Enabling this complex sector to compute with uncertainty to estimate financials could improve access to funding and help other nascent industries.

**Keywords:** financial risk assessment; vertical farming; urban agriculture; probability bounds analysis; economic viability

## 1. Introduction

Agriculture faces a plethora of threats including unusual weather phenomena, water shortages and ageing rural populations [1]. These combined challenges require innovation in resilient farming methods to meet the demands of a growing population. Vertical farming (VF) is one such method that may contribute towards food and nutritional security.

VF is a novel form of agriculture, defined as multi-layer indoor crop production systems with artificial lighting, in which growth conditions are controlled [2]. Plants can be stacked vertically (in towers) or horizontally (in trays or gullies) [2]. The goal is simple, to produce more food with less land. It utilises controlled-environment agriculture (CEA) techniques, such as hydroponics with growing-specific light-emitting diodes (LEDs). Figure 1 maps the spectrum of agricultural systems across two gradients in technology and exposure to nature.

Indoor vertical farms, otherwise known as plant factories with artificial lighting (PFALs) [1], are typically the most technology-intensive and expensive. Consequently, they can control most growing parameters independently of external environment factors. This unprecedented level of control has enabled research to optimise production by fine-tuning variables, including light spectrum, temperature, and irrigation [3,4]. With such control,

VF offers a host of advantages when appropriately managed, including higher yields all year round, quicker feedback cycles, longer shelf-life, and zero pesticide usage [1]. This form of agriculture can utilise the internet-of-things and big data to achieve smart factory performance [5]. The most popular crops to farm vertically are leafy greens, herbs, and microgreens due to high energy conversion to edible matter. Technically it is possible to grow any crop; however, economics and growing complexity constrain crop choice.

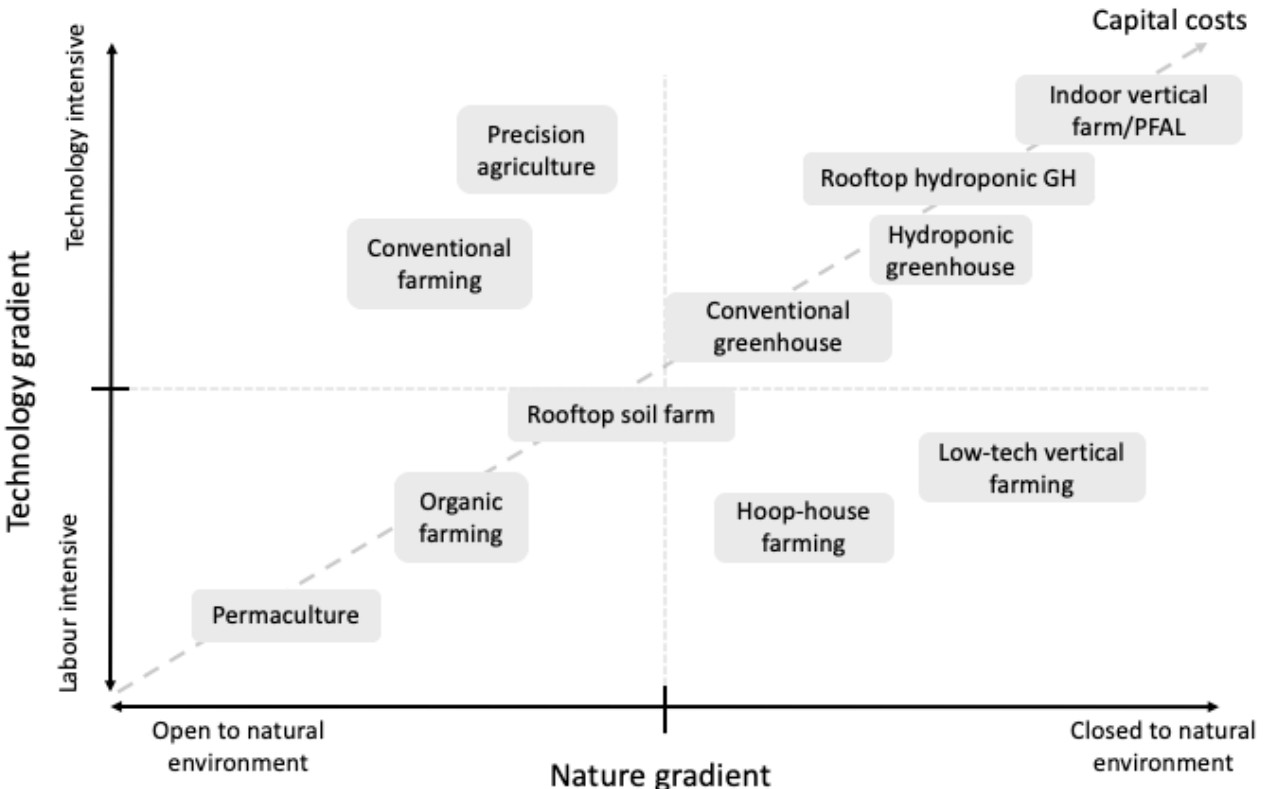

**Figure 1.** Spectrum of farm types (adapted with permission from C. Peterson & S. Valle de Souza [6]). Capital costs increase the further away a farm type is from the bottom left.

The industry has seen a surge of interest and significant investments in recent years [7–9], driven by advances in light-emitting diode (LED) technologies over the past decade. As a result, vertical farms are sprouting up worldwide, particularly in locations that make strategic sense (environments hostile to crops, regions with cheap electricity and markets for premium-quality food). The practice is not widespread and attracts scepticism. Criticism is focused on high capital and operational costs due to expensive equipment and the high-level expertise required to operate it, and high energy demands, which can result in low profit margins [2,10,11]. The learning curve is steep as the market, expertise, and technology begin to mature.

Market drivers are in VF's favour; however, there have been numerous failures over the past decade [12]. Continued investment is usually needed to sustain vertical farms; otherwise, they may bleed dry from negative cash flow [13,14]. Therefore, there remains hesitance to invest in VF [12,15]. A recurring complaint from investors, researchers, and practitioners is the scarcity of peer-reviewed research investigating economics underlying the construction and operation of VF [6,16–18]. Despite vertical farms operating in controlled environments and utilising data to optimise growing conditions, there is a lack of production, yield, and economic data available in the literature [12,18]. This is amplified by the absence of any standardised data framework and benchmarking. Variations in data quality due to complex climate controls and differing technologies, sensors, and yield measurement practices mean that data are not always applicable across farms. There

are industry working groups now working towards standardisation [19,20]. The void of validated and peer-reviewed economic and risk data in the literature highlights a vital need for addressing the economics of VF so that it can be improved. One way to circumvent this is the utilisation of risk and uncertainty quantification techniques. In principle, risk management would reduce profit fluctuations and increase investments whilst raising farmers' income. As a consequence, improved access to finance could help with achieving sustainable development goals [21].

VF is a high-risk business, yet no efforts have been made to quantify and evaluate financial risk in the literature. There is a need to factor risk and uncertainty into business models for a more accurate assessment and to increase accessibility to funding [22]. This article explores whether VF economics can be analysed through a novel economic risk methodology, allowing imprecise random variables to assist farm owners and investors in making financially sensitive decisions. It aims to address the following research questions:

- How can farm economics be modelled with an absence of available production, risk and financial data to conduct economic viability and risk assessment?
- What is the risk profile for two case study farms, one of which benefits from a synergistic partnership with a landlord and cost deductions?
- How might a risk assessment tool be used to inform a profitable business model?
- The article is structured as follows:
- In Section 2, related works and their inability to accurately assess the economic viability of VF projects are discussed alongside potential risks;
- In Section 3, the model is proposed alongside the risk and uncertainty quantification methods, as well as the two case study farms;
- In Section 4, the results from the analysis are presented for financial metrics;
- In Section 5, the results are discussed alongside possible interventions to de-risk one case study, the implications of using the methods proposed in the broader industry, and the limitations of the analysis are discussed; and
- In Section 6, the conclusions are presented.

## 2. Related Works

In this section, the related works on the economics and risks of VF is investigated. Economic models on VF are grouped and then examined for their insights and challenges. Typical risks of the sector from VF and CEA are described.

### 2.1. Economic Analyses

There are 16 disparate economic analyses from academic and commercial sources detailed in Table 1. The literature reflects the nascence of the industry.

**Table 1.** Vertical farming economic analyses alongside their characteristics.

| Type | Source | Objective | Results |
|---|---|---|---|
| Cost analyses | [23] | Simulate the economics for a hypothetical 37-storey (167.5 m) vertical farm hybrid in Berlin, Germany. | Cost of production presented through probability distributions. Costs lie between €3.5–4 per kg in 44% of cases. No validation. |
| | [24] | Simulate life cycle costing for a hypothetical 50 m$^2$ apartment to study small and inexpensive VF. | Sensitivity analysis results indicate added value crops such as herbs and pharmaceutical ingredients are necessary for economic viability. No validation. |
| | [22] | Provide a business planning spreadsheet developed for a hypothetical 1000 m$^2$ PFAL based on expert's and industry practitioners' insights. Most comprehensive data set in the literature. | Cashflow projections for a profitable farm with a 7.8 payback period. |

**Table 1.** *Cont.*

| Type | Source | Objective | Results |
|---|---|---|---|
| | [25] | Conduct feasibility study using central limit theorem to assess ROI for a hypothetical 5000 m$^2$ VF serving 24 canteens in Wuhan, China. | The breakeven on investment in this VF analysis is 11.5 years. Unviable crops are selected. |
| | [26] | Perform cost analysis for a hypothetical ZipGrow VF in São Paulo, Brazil comparing to Denver, North America, assessing its economic viability using vendor's data. | São Paulo provides a cheaper scenario in comparison to Denver, but possesses market conditions where low costs cannot compete with traditional farming product prices. Analysis predicts Denver as 14.17% IRR compared to −19.12% in Sao Paulo. |
| | [27] | Analyse the economics of a hypothetical six-story VF in Delhi, India, with a footprint of 200 m$^2$ and 3 stacked layers in each story. | Payback period calculated to be 64 years. Unviable crops are selected. |
| | [18] | Draw from hypothetical Japanese PFAL data [22] and substitute modern data in various scenarios (changes to scale, operations and market context). | Significant decline in capital costs, especially equipment (45%), make profitability increase substantially (ROI rose from 1.8% to 14.3%). Scale of operation is critical to profit as well and depends on the proportion of fixed costs in the operating structure. Doubling the size of the PFAL results in the enhancement of ROI from 14.3% to 22%. |
| Software systems | [28] | A flexible system for predicting costs and return-on the investment of a VF, with results shown for several hypothetical scenarios and sensitivity analysis. | Return on investment is sensitive to price of electricity, crop price and $CO_2$ concentrations. Software not publicly available. |
| | [29] | A commercial and flexible digital platform for economic estimation of farms, greenhouses and VF. | Capital expenditure, operating costs and yield estimates alongside 15-year projection. Not peer-reviewed or academically validated. |
| | [12] | Evaluate business sustainability using imprecise data techniques using ideas from [28]. The economic modelling contained within "How High is High Enough?" builds upon the framework and executes the first passage time risk analysis on two case studies. | N/A—No results presented. |
| Greenhouse vs. VF | [30] | Simulate a hypothetical scenario comparing profitability of growing lettuce in a semiclosed VF and semiclosed GH farm near Quebec City. | Results show that the costs to equip and run the two facilities are similar with higher gross profit for VF. |
| | [31] | Simulate scenarios to compare hypothetical VF and GH facilities under various financing schemes in Denmark. | Results show that regardless of financing scheme, the VF facility was much more profitable compared to the GH, with high IRR rates and a payback period between 2–6 years. |
| Industry surveys and reports | [32] | Present results of a self-reported survey of 56 indoor VFs (primarily in the USA). | Aggregated data for OpEx breakdowns per and profitable crops |
| | [11] | Present results of the government census of a number of profitable Japanese plant factories with typical production costs. | Aggregated data for production costs and percentage of profitable farms in Japan. |
| | [33] | Present results of a self-reported survey of 190 indoor VFs. | Aggregated and self-reported data on profitability and revenue. |
| | [34] | Design and cost an economically feasible next-generation VF concept. A workshop of experts design and cost five hypothetical food modules with margins to account for uncertainty. | The resulting concept is broken down into estimated capital expenditure and running costs. |

Records and financial data on vertical farms are scarce, and this is demonstrated by the fact that most of the analyses are based on hypothetical case studies. The farms in these studies range from skyscrapers [23,25,27] to more realistic warehouses [35] and small-scale operations [24,31]. The sector has been notorious for being closed, yet it is starting to shift due to the immense complexity of combining elements of lighting, plant science, engineering, policy, architecture, and sustainability [19,36]. Currently, VF studies

commonly extrapolate data from greenhouse literature [28,30,31], estimate values [23] or utilise projections from vendors [26,37].

Cost Analyses and Scenario Simulation

These analyses discuss the categories of capital expenditure (CapEx) and operational expenditure (OpEx) alongside the methods used to compute productivity and profitability [22–27,34]. Most of these struggle to provide a balanced assessment of feasibility of the VF projects due to an absence of empirical data. The complex nature of combining architecture, agriculture and digital technologies in an urban food-water-energy nexus context makes accounting difficult. The most comprehensive dataset of a vertical farm is a hypothetical PFAL in Japan [22]. One recent study expands on this dataset to test various scenarios with an updated capital cost reduced by 45% due reduction in equipment costs (changes to scale, operations and market contexts) [18]. It reveals that doubling the production scale with the same fixed costs can increase the return on investment from 14.3% to 21.7% [18]. Moreover, profitability hinges on commanding a premium price point whilst reducing costs (such as electricity through LED efficiency) without sacrificing produce quality [18]. It concludes that scale of operation, reduction in capital cost, and innovations in improving yield and produce quality are critical to profitability [18].

Economic Estimation Software

Customisable analyses are necessary to accommodate various scenarios and user inputs, especially as datasets are hard to come by. Tools exist that aim to help entrepreneurs compare different locations, systems, and business models [12,28], but only one is available for commercial use [29]. As a commercial tool, it lacks the rigour of peer-reviewed yield values and does not currently allow the user to consider any uncertainty or risks. Moreover, it is a black box and is therefore challenging to critique; [28] is not fully functional but the model informed [12], which provides the framework executed within this study.

Greenhouses vs. Vertical Farms

There are mistakes that can easily result from hypothetical data. Two studies conclude that vertical farms are more profitable than greenhouses in certain conditions [30,31]. Upon closer examination, the values for space utilisations (defined as floor space dedicated to growing divided by facility area) are unfairly skewed in favour of VF for both studies. Space utilisations are typically 50% for VF [11] and 60–90% for greenhouses [38]. Thus, the studies are misrepresentative of real farms. If an analyst adjusts the space utilisations to realistic values, then greenhouses are more competitive then the results suggest. If it were possible to compute with uncertainty about these assumptions, then perhaps false conclusions could be avoided. Neglecting depreciation is another critical mistake, as a comparison study claims that vertical farms are more profitable [30] without consideration for depreciation of vertical farming equipment like lighting. Greenhouses may use supplemental lighting but they are not in-use for up to 16 h a day all year, and therefore depreciation will happen at a much slower rate compared to VF.

Industry Surveys and Reports

These are the three analyses utilising real-life farm data, albeit two are self-reported surveys without auditing and are aggregated across different farm types, making them difficult to compare [32,33]. Nevertheless, they collectively cover a dataset of 461 vertical farms and provide some overview statistics including the percentage of profitable vertical farms increasing each year [37]. Some also include the percentages of cost components [11,32] and a snapshot of the average labour (0.0155–0.03 people per square metre) and water required (an average of 1.69 litres per square metre) [32].

2.1.1. Cost Components

Three elements primarily drive CapEx comprising 80–90% of costs: lighting, racking and grow system, and building [37]. The production costs consist of three major constituents that account for 75–80%: electricity, labour and depreciation [35,37]. There is no

analysis whereby all cost components are considered. To highlight the disparity between both the real-life and hypothetical data for OpEx and CapEx, [37] collates all the available information for fixed and variable costs. This collation shows that researchers frequently omit heating, ventilation and air-cooling (HVAC), depreciation and $CO_2$ enrichment. Resource data are speculative in most cases.

### 2.1.2. Uncertainty

To date, most of the analyses rely upon deterministic models to predict cash-flows [26,27,29,30,34,35,39,40]. Scarce data have forced researchers to utilise uncertainty quantification techniques in order to bolster analyses and improve accuracy [24,25,28]. World-leading researchers in plant factories claim that a risk scenario approach would benefit the sector but would require industry-wide research and cooperation (involving horticultural scientists, farm operators, equipment manufacturers, etc.) [35]

Stochastic methods are utilised in several models, such as central limit theorem [25], scenario analysis [23,31], sensitivity analysis [24,28] and probability bounds analysis [12]. Sensitivity analyses determine that profitability is sensitive to electricity price, crop price, sunlight contribution, photosynthetic photon flux density, and LED fixture efficacy [24,28]. These factors highlight the importance of electrical efficiency and suitable sales models.

### 2.1.3. Limitations

The primary source of error is that many of these analyses utilise speculative assumptions without accommodating uncertain inputs. An attempt to calculate uncertainty would represent more realistic cash flow predictions, especially as projected yields and costs can be misrepresentative [14]. Researchers often overlook HVAC costs in most economic analyses due to their complexity. Additionally, labour is costly, and automation solutions like seeding machines, packaging machines, and nutrient delivery systems are popular solutions, yet no analyses consider automated systems in their cost breakdowns. Researchers and industry practitioners recognise the need for more detailed economic analysis that model all the variable costs to inform business models and financial investment [18,23,28,41]. Without this and the lack of proven business models, there is insufficient evidence to address criticisms regarding profitability. Moreover, all of the analyses are for unique farms and production systems with differing levels of technology and operating with different economies, making performance not directly comparable.

The learning curve is a vital element considered in only two cases [12,29]. Farms can experience an improvement in yield and produce quality depending on growing experience, wastage and the optimisation of parameters [29]. This improvement should be tracked in future studies for validation.

No studies have addressed the fundamentals of microeconomics, such as maximising profit and average cost curves. This would enable the assessment of economies of scale and finding the 'sweet-spot' in terms of facility sizing. Access to real data would reduce epistemic uncertainty in analyses. A credible foundation for literature will then develop. Computational uncertainty quantification could compensate for lack of available data. Lastly, risks and opportunities can be applied. A tool that could achieve this can inform decision-makers of VF viability with confidence and avoid costly failures. Other limitations are discussed within a review [37].

### 2.2. Risks and Opportunities

The VF sector is littered with failed start-ups, some of which have been spoken about publicly [42,43] and many that go unreported. Reasons for ceasing trading include:
1.      cash flow problems [14,44];
2.      underestimated labour costs due to operational complexity [14,42,45];
3.      lack of adequate knowledge and accessible education about the integration and operation of vertical farming systems (irrigation, lighting, plant science, HVAC and manufacturing systems) [14,42];

4.  inefficient workflow and inadequate ergonomic design consideration [14,42,46];
5.  low profitability margins [46];
6.  sources of capital investment and the misalignment of support and expectations from funders [42];
7.  zoning codes and regulatory obstacles [14,47];
8.  equipment failures and associated repair costs [42,48]; and
9.  poor early decisions around pricing, crop selection and location [12,42,49,50].

These failures are acute because of the high CapEx investments required. The economic analyses omit all these risks that may influence crop productivity, sales, and profitability [14]. No empirical data exists for the frequency and impact of such events in VF except for anecdotal reports [14]. On the other hand, the literature on risk analysis in greenhouses and field-grown agriculture is more mature [51–59]. The sources of risk range widely. As indoor farming climbs the technology and nature gradient (see Figure 1) its risks shift away from external environmental factors and towards production risks associated with technology. Table 2 identifies and ranks the likelihood for risks for field-grown produce, greenhouses from the literature and compares against vertical farms based on anecdotal reports [14,42].

**Table 2.** Risk identification and corresponding likelihood for vertical farm, greenhouse and field-grown produce (cf. [14,51,57,60–62].)

| Risk Parameters | Risk Source | Indoor Vertical Farm | Greenhouse | Field-Grown |
|---|---|---|---|---|
| Yield risk | Weather conditions | Low | Medium | High |
| | Pest outbreak | Low | Medium | High |
| | Pathogen outbreak | Medium | Low | High |
| Production risk | Environmental control (malfunctioning HVAC) | High | Medium | Low |
| | Electrical outage | Medium | Low | Low |
| | Incorrect nutrient/pH dosage | Medium | Low-Medium | Low |
| | Irrigation (flooding, clogs) | High | Medium | Low |
| | Equipment failure | High | Medium | Low |
| Cost risk | Energy expense variability | Very High | High | Low |
| | Underestimated labour costs | High | Medium | Low |
| | Technology advances | High | Medium | Low |
| Labour risk | Poaching of staff/Loss of expertise | High | Medium | Low |
| | Accidental damage | High | Medium | Low |
| Safety risk | Fire | Low | Low | Low |
| Planning risk | Zoning codes | High | Medium | Low |
| | Change of lease agreement | High | Medium | Low |
| Market risk | Market competition | Medium | Medium | Low |
| | Local supply/demand situation | Low-Medium | Low | High |

Economists model such risks according to probability distribution functions known to decision-makers [57]. However, in empirical analyses, researchers almost never know the true probability distributions [57]. Economists assume that decision-makers hold beliefs consistent with known probability distribution functions. Rather than assuming the exact distribution whilst lacking adequate data, imprecise data techniques are better suited for estimating this.

Innovations in the VF sector have arisen to address the challenges and improve unit economics in an increasingly competitive market. Therefore integrating opportunities are equally important to consider. PFALs in Japan report that cost performance can be radically improved by reducing production costs and increasing annual sales [35]:

- A 50% increase in sales is achievable within five years by adjusting environmental control setpoints, selecting better cultivars, improving the cultivation system and reducing waste [35].
- A 50% reduction in production cost is possible through improving labour and electrical efficiency [35]
  - Automation, process flow and human resource development can reduce labour costs.
  - A 50% reduction in electrical cost is attainable within several years through the intelligent operation of electrical systems, insulation, LED efficiency advancements [35] and load shifting [63].

Other opportunities such as new customer contracts, introducing new technologies and scaling plans are out of the scope of this article.

## 3. Methodology

This methodology is broken down into several sections:

1. The economic model containing its framework and assumptions to calculate cashflow forecasts and return on investment (ROI);
2. The risk and uncertainty analysis, which describes the methods used, why they were used, the risk profiling results and the risks that will be considered within this analysis;
3. The case studies and associated data for a real-life and hypothetical farm.

### 3.1. Economic Model

The economic survivability model is a flexible and robust means to conduct financial risk assessment by combining historical data with risk and uncertainty quantification to fill gaps in knowledge. This method is based on previous work [12]. The model functions through a series of modules that interprets inputs based on the local market, selected crops, farm characteristics, labour, consumables and more. The flow of tasks is illustrated in Figure 2.

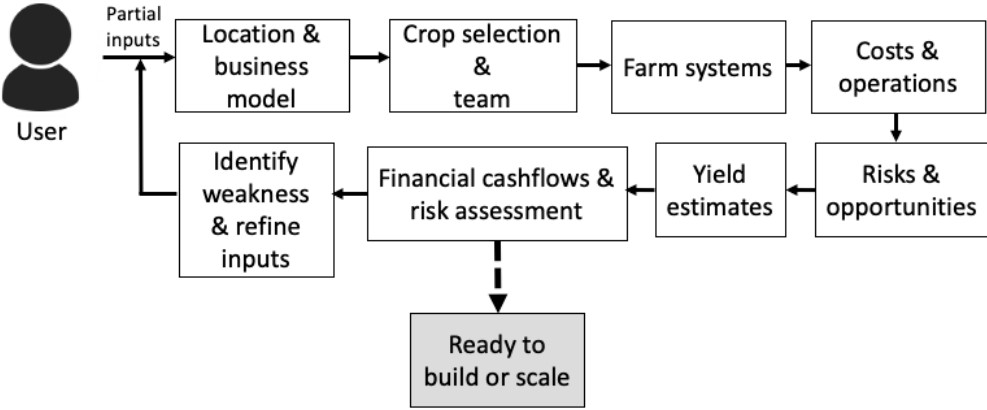

**Figure 2.** Flow chart of user interactions with model.

The model computes cashflow forecasts and ROI based on either farm inputs or default values. Default values are estimated by decisions on location, system selection, crop type, farm size and other inputs based on the literature [12]. Once the inputs have been gathered, risk analysis is conducted using first-hitting-time, which will evaluate whether the farm is likely to fall under certain criteria in the future when accommodating for risks as well as reported opportunities. The novel application of probability bounds analysis enables the use of both complete and partial inputs where the specified farm (in planning or operational stage) does not have complete information.

Figure 3 shows the simplified flow of computation and cost components from left to right, whilst omitting the interdependencies inherent in plant growth. The model calculates

revenues and costs such as CapEx, OpEx and cost of goods sold (COGS) for resulting ROI. To illustrate how the model functions to compute risk profiling, Figure 3 is labelled with numbers 1 to 11 corresponding to equations available within the Supplementary Method Statement. This information is collected through a series of spreadsheets before being processed by a Python script to apply uncertainty quantification and produce cashflows with risk profiles for quasi-insolvency. This is applied across all the potential scenarios based on user uncertainties, risks, and opportunities, relevant to the farm type. The resulting analysis is a 15-year projection for financial metrics and resource consumption, as the typical lifetime for a vertical farm is approximately 15 years [11].

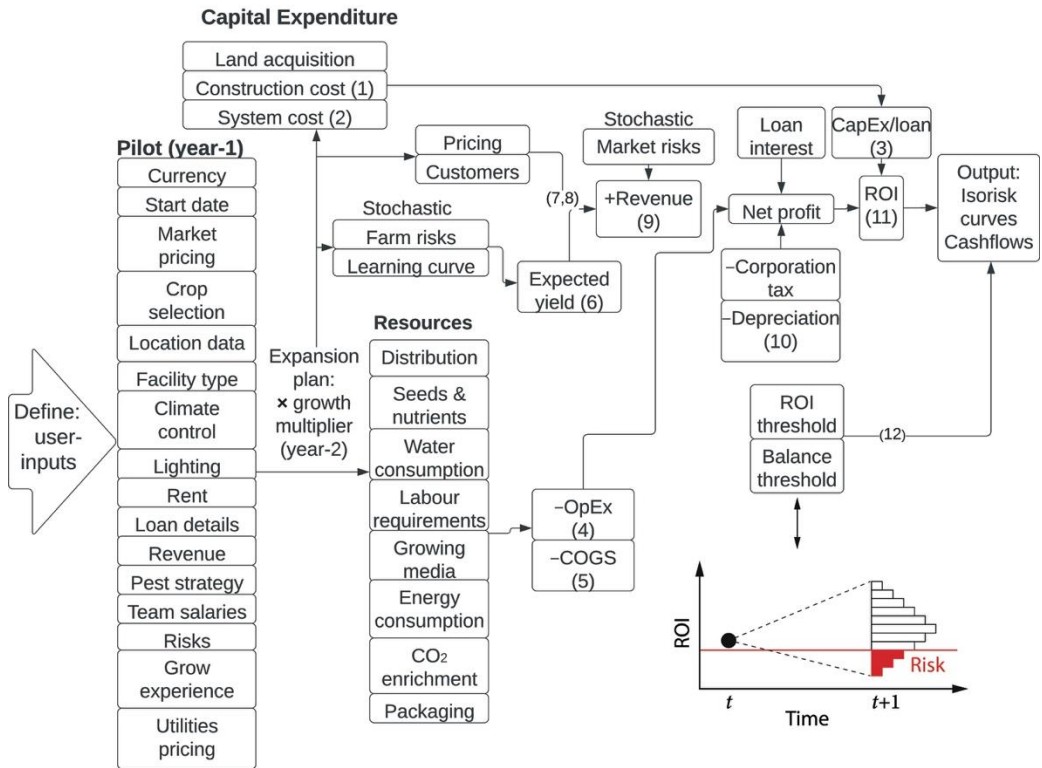

**Figure 3.** Financial risk model structure (flow left to right) utilising Equations (S1)–(S12).

Refer to Supplementary Method Statement for detailed breakdown of the model including its equations, assumptions and references.

*3.2. Risk and Uncertainty Analysis*

Stochasticity is included through random parameters such as failure rate, improved yields over time, repairs, infrastructural issues, potential pest or pathogen outbreaks and other risks. The user can also manually insert uncertainty for any parameter. How can these be accounted for if the distributions and values are unknown? Probability bounds can capture all information, even if there is only limited information available.

Probability bounds, expressed as bounds on cumulative distribution functions, are called "p-boxes" [64]. They can be used to characterise uncertain parameters, distributions, risks and opportunities without requiring overly precise assumptions [65]. There were other uncertainty techniques that could have been used instead, like Monte Carlo simulation or worst case analysis. However, this would require untenable assumptions, such as the uncertainties being small, the distribution shapes are known and the relevant science is modelled [66].

This is not the case, and p-boxes can overcome these limitations through using all the information available (even if partial) without making over-simplified assumptions. Figure 4 shows how imperfect information may be presented in a p-box form on a cumula-

tive distribution function (CDF) whereby A's distribution is known, but not its parameters, B's parameters are known, but not its shape, C has a small empirical dataset, and D is known to be a precise distribution.

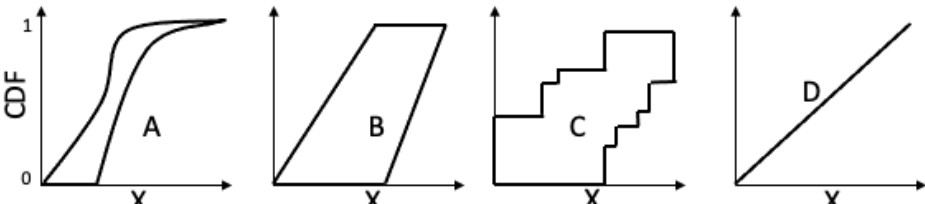

**Figure 4.** Probability boxes representing different types of uncertainty (cf. [66]).

The integration of probably bounds analysis enables model inputs with partial information such as an input interval of 30–50 h of direct labour per week (expressed as an min-max interval '30,50'). Moreover, the probability of a pest outbreak occurrence in a given year might be between 35–70% with a single best estimate of 50% (min-max-mean '30,70,50'), with the associated impact being 0–25% of annual yield conveyed as a beta distribution. A breakdown of the risks and their weighting according to model parameters is included within the method statement and found within 'risk_pba.py' within the Model Library in Supplementary Materials. The central limit theorem may be incorporated to give a yield estimate using a normal distribution rather than a precise value [25]. This approach accounts for risks and opportunities that would be nonsensical to provide a precise probability or impact without any historical or peer-reviewed data. In this analysis, the 'pba' package on Python [67,68] was extended to execute the probability bounds analysis necessary.

Once p-boxes are integrated within the model and a simulation has been executed, the resulting finances are analysed. The probability of the cashflows and projected ROI falling below a 'bankruptcy' threshold can be used to predict the event of insolvency defined as the first-hitting-time. First-hitting-time is a method used commonly to predict 'survival' in economics [69,70] and other disciplines [71–73]. This hybrid approach of p-boxes with first-passage time has only been applied in one instance for calculating ecological extinction risk [72], and would allow the assessment of financial risk despite deep uncertainty. As historical data and refined inputs are added, the p-box would shrink in size to compute more precise risk-profiling and financial projections.

The quasi-insolvency thresholds are defined as cashflow becoming negative ($T_B$) and an ROI under a threshold specified by the user ($T_{ROI}$). Based on a review of bankruptcy models that evaluated whether the most important and frequently used financial ratios are within the profitability group [74], this analysis focuses on the profitability metrics to assess insolvency. The company under analysis is at risk of insolvency when they have no capital runway, which means they will collapse if they do not raise additional capital whilst their revenues and expenses remain unchanged. For ROI, a venture capitalist would typically look for a return of 10–20%+ [46]. The threshold for ROI may vary with time according to investor demands. The probability of insolvency for a given year (INS) is therefore defined in Equation (1).

$$P(INS) = P[\,(B < T_B)\ \&\ (ROI < T_{ROI})]$$ (1)

The p-box represents all the possible scenarios modelled and the probabilities of insolvency. The resulting risk analysis can be made useful by introducing categories defined by probability of insolvency over some defined time scale:

- *Critical:* 50% probability of insolvency within 3 years
- *Substantial risk:* 25% probability of insolvency within 5 years
- *Moderate risk:* 10% probability of insolvency within 10 years
- *Safe:* Less than 10% probability of insolvency within 10 years

These categories are mapped onto the analysis to communicate the level of uncertainty and risk profile of the farm. Figure 5 shows an example of the risk assessment. The p-box (shaded in grey) primarily falls within the moderate risk category with some creep into safe and critical due to a large degree of uncertainty. This highlights a lack of either precise inputs or information about impacts and the frequency of risks. The future is unknown, but with risk mitigation and corrective action the risk profile could be improved.

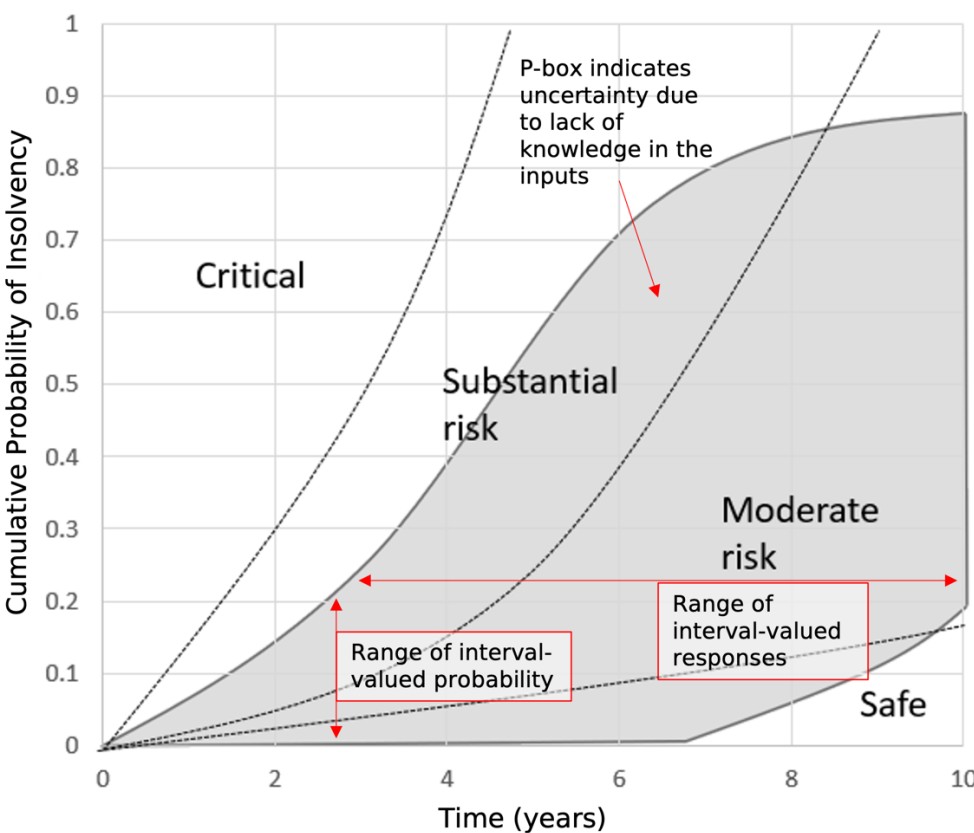

**Figure 5.** A risk curve using probability bounds (shaded in grey) and first-hitting time to evaluate the risk profile of a VF insolvency.

### 3.3. Farm Case Study Inputs and Assumptions

Two vertical farm case studies are used for this analysis: a real commercial vertical farm based in the UK and a hypothetical vertical farm in Japan informed from the literature [35]. The data for the UK case study is for a small-scale commercial VF and has been collected on-site. The information for the Japanese farm is a complete business plan example available within the literature based on the real-world experience of twenty scientists and business managers in the sector [22]. Both examples have been selected because their crop choice of leafy greens is the dominant cultivar in this sector [75]. The methodology described will be applied to both case studies in order to evaluate their profitability and risk profiles. The assumptions about the farm are listed in Table 3.

**Table 3.** Assumptions for UK and Japanese case studies (cf. [22]).

| UK Vertical Farm | Japanese Vertical Farm |
|---|---|
| 1. The farm has been retro-fitted and installed into a basement rented from a school. The school subsidises rent, electricity and water costs. | 1. The farm has been constructed within a leased purpose-built facility. |
| 2. The facility is a pilot with plans to double production capacity in the next year. Therefore, the analysis considers both the pilot and full-scale plan. | 2. The facility is at full production capacity with no plan to expand. |
| 3. Vertical towers were modelled as a growing area. The farm's imprecise yield data are used to form upper and lower bounds to compensate for the lack of robust data collection. | 3. Nutrient Film Technique (NFT) racks were modelled with the annual yield provided in the example. |
| 4. Lettuce cultivars are grown with twelve plants per tower and a growth cycle of 21 days (after 25 days in the propagation system). | 4. Lettuce heads are cultivated in four phases at different spacing: 1st seedling (8 days), 2nd seedling (10 days), transplanting 1st (8 days), transplanting 2nd (8 days). |
| 5. Alternative revenue streams (such as education) are omitted to assess the farm in isolation. | 5. No alternative revenue streams are included. |
| 6. Water consumption data are tracked on the farm for 15 months and have been characterised per month: min = 1325 L, max = 8325 L, mean = 3730 L, Standard deviation = 2039 L. Multiplied by 2 for the scaled-up plan. | 6. Water costs have been grouped with electricity costs. |
| 7. The facility has a pre-existing HVAC system that has no associated capital costs. | 7. A bespoke HVAC system was installed. |
| 8. The indirect team consists of three staff (head grower, marketer, manager). | 8. Indirect staff costs were not considered by [22]. This analysis assumes five staff members (CEO, head grower, marketer, engineer and admin). |
| 9. The farm is partly grant-funded for two years. | 9. The project is funded with zero interest rates, according to [22]. |
| 10. The farm is partially insulated within a thick-brick walled basement but is not sealed, which reduces the climate control capacity. | 10. The facility is insulated and benefits from a strictly controlled environment. |

A summary of characteristics for the scaled-up UK farm and the hypothetical Japanese are given in Table 4. Then, a capital cost breakdown (Table 5) is followed by an operational cost breakdown (Table 6). All inputs can be found in the Supplementary Data, Tables S10 and S15. All values are converted to GBP with a conversion rate of 1 USD = 0.72 GBP.

**Table 4.** Farm characteristics summary for UK and Japanese farms (adapted with permission from [22]).

| Characteristic | UK Farm | Japanese Farm | Unit |
|---|---|---|---|
| **Real Estate** | | | |
| Facility size | 220 | 1000 | $m^2$ |
| Facility height | 3 | 3.5 | m |
| Space utilisation | 45 | 36.4 | % |
| Growing space | 100 | 364 | $m^2$ |
| **Systems** | | | |
| Grow levels | 30 towers per rack | 6 shelves | |
| Number. of racks | 16 | 241 | |
| Stacked growing area | 392 | 2184 | $m^2$ |
| Number of lights | 256 | 5784 | |
| Light wattage | 100 | 32 | W |
| Energy price | 0.073–0.108 | 0.090–0.100 | £/kWh |
| Annual electrical consumption | 224,255 | 1,676,052 | kWh |

**Table 4.** *Cont.*

| Characteristic | UK Farm | Japanese Farm | Unit |
|---|---|---|---|
| **Labour** | | | |
| Number of direct labourers | 3 | 9 | people |
| Number of indirect staff | 3 | 5 | people |
| Direct labour hours per week | 20 | 42 | hours per person |
| Direct hourly cost | 9.50 | 7.34 | £/hour |
| **Crop: Lettuce** | | | |
| Annual yield | 8800–10,800 | 116,640 | kg/year |
| Harvest weight | 0.1 | 0.09 | kg |
| Photoperiod | 16 | 16 | hours |
| Product weight | 0.3 | 1 | kg |
| Customer segmentation | 85 (customer 1) 15 (customer 2) | 100 | % to customers |
| Unit prices | 7.50 (customer 1) 3 (customer 2) | 8.64 | £/unit |
| Packaging cost | 0.85 | 0.05 | £/unit |
| **Attributes** [1] | | | |
| Business model | Hybrid | Wholesale | |
| Grower experience | Medium | High | |
| Automation level | None | Medium | |
| Climate control level | Medium | High | |
| Lighting control level | Medium | High | |
| Nutrient control level | Medium | High | |
| $CO_2$ enrichment | No | Yes | |
| Biosecurity level | Medium | High | |

[1] Definition of input is detailed in method statement in the Supplementary Materials.

**Table 5.** Capital cost breakdown for full-scale UK and Japanese farms (adapted with permission from [22]).

| Capital Costs | UK Farm | Japanese Farm | Unit |
|---|---|---|---|
| **Construction** | | | |
| Finishing | 3850 | 114,775 | £ |
| Appliance | 4250 | 108,000 | £ |
| Management costs | 9029 | 0 | £ |
| Electrical infrastructure | 8020 | 25,200 | £ |
| Real estate | 0 | 0 | £ |
| Total construction costs | 25,149 | 247,975 | £ |
| **Systems** | | | |
| Growing system cost | 55,071 | 747,072 | £ |
| Lighting system cost | 87,165 | 538,804 | £ |
| HVAC system cost | 2700 | 56,160 | £ |
| Miscellaneous cost | 9548 | 0 | £ |
| Total equipment cost | 154,484 | 1,342,037 | £ |
| Total capital costs | 179,633 | 1,590,012 | £ |

**Table 6.** Operational costs breakdown for the full scale UK and Japanese farms (adapted with permission from [22]).

| Production Costs | UK Farm | Japanese Farm | Unit |
|---|---|---|---|
| **Operational expenditure** | | | |
| Rent | 0 | 69,120 | £/year |
| Staff costs (non-direct labour) | 70,236 | 171,888 [1] | £/year |
| Distribution | 31,172 | 106,691 | £/year |
| Other costs [1] | 1404–6039 | 8594 [1] | £/year |
| Total OpEx | 108,998 | 356,293 | £/year |
| **Cost of goods sold** | | | |
| Direct labour costs | 29,640 | 142,689 | £/year |
| Growing media | 5735 | 14,818 | £/year |
| Packaging | 22,977–32,078 | 2905 | £/year |
| Total electricity cost | 15,929–23,416 | 150,844 | £/year |
| Water cost | 97.59 | N/A | £/year |
| Total COGS | 104,000 | 375,192 | £/year |
| Other costs | | | |
| Depreciation | 20,417 | 162,454 [1] | £/year |
| Working capital | 251,504 | 2,160,000 | £ |
| Loan amount | 158,000 | 0 | £ |
| Loan tenure | 7 | 0 | years |
| Loan interest | 5 | 0 | % per year |

[1] Inputs have been modelled based on assumptions in absence of data.

## 4. Results

The case study business scenarios (in Section 3.3) are simulated over a 15-year period, the typical lifetime of a vertical farm [11], for cash flows and financial risk analysis. They enable the evaluation of economic viability. The graphical results depict the lower bound on the 2.5th percentile (labelled as 'Min'), the upper bound on the 97.5th percentile ('Max'), the lower and upper bounds on the median (labelled as 'Lower Median' and 'Upper Median') of each variable of interest. The median provides insight into the value at which 50% of all the possible scenarios are above or below.

Each case study will include financial balance, annual yield, return on investment and risk assessment. Two of these metrics, financial balance and return on investment, are used to compute the risk of insolvency and therefore include a threshold. In this analysis, the risk is defined as the combination of negative cash flow and underperforming ROI, which is characterised by probability. The cumulative probability of both of these metrics falling under their respective thresholds simultaneously dictates the risk visualised. The model can easily be generalised for other financial metrics or definitions of risk. Other financial metrics and their respective max–min cases considering with and without risks and opportunities are presented in the Supplementary Data in Section 1.5 (UK farm), 1.7 (UK farm post-intervention), and 2.5 (Japanese PFAL). The full results can also be found as 'results_UK.py', 'results_UK_post.xlsx' and 'results_JPFA.xlsx' for the UK farm, UK farm post-interventions and Japanese farm respectively within the Model Library in Supplementary Materials.

### 4.1. UK Vertical Farm

The UK small-scale farm begins its operations with a financial balance of £180,000, which is projected over the 15-year period (see Figure 6) with increasing uncertainty. 50% of the scenarios represented by the median are split above and below the risk threshold.

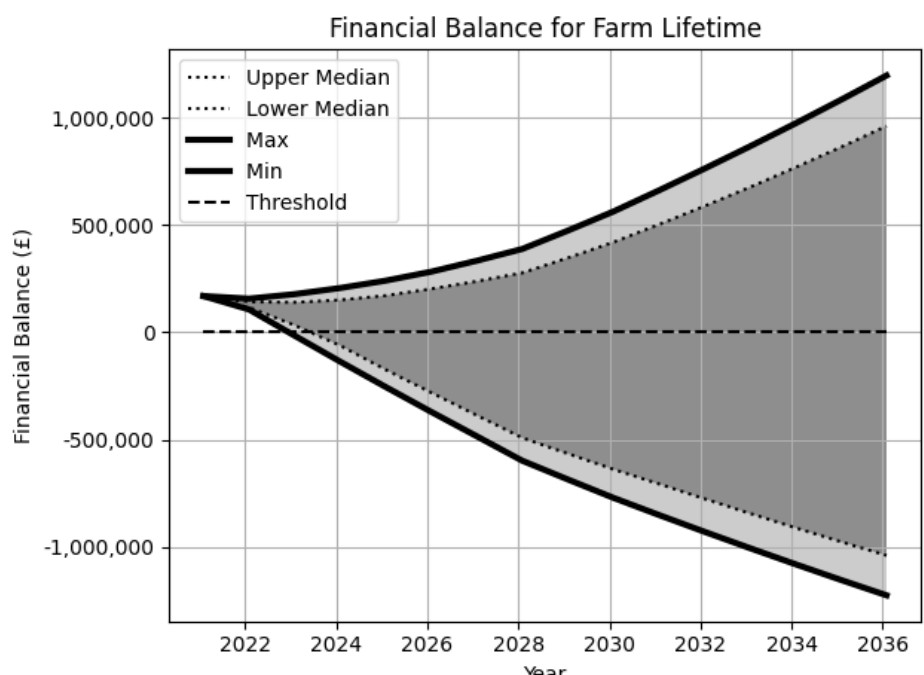

**Figure 6.** Uncertainty about financial balance for the UK farm over the 15-year simulation.

The annual yield for the UK farm for lettuce production is shown in Figure 7. There is a sudden increase in yield as the farm scales to full production (doubling the amount of growing systems in the facility) in 2023. There is also a high degree of uncertainty due to the lack of accurate yield tracking on the farm and the possible effects of pathogens and pests. The median is large due to input uncertainty without statistical data such as light efficiency improvements and electricity price. The effect of reducing waste and improving yield as the farm staff gain experience is reflected in the positively increasing gradient of both the max and min scenarios.

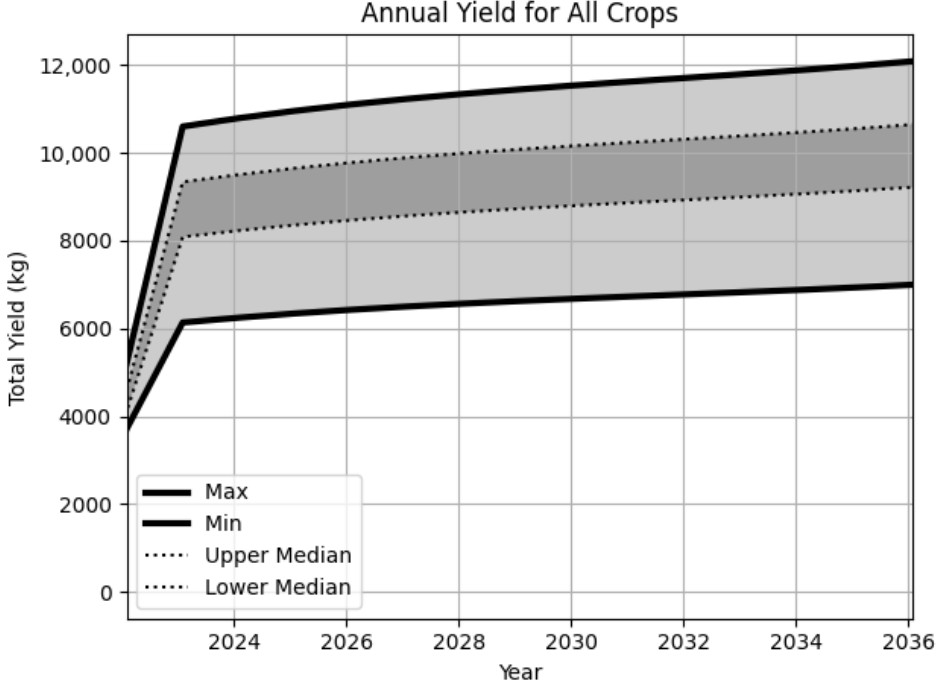

**Figure 7.** The annual yield for the UK farm has a range between 6000 kg and 11,000 kg after scaling up in 2023. The median annual yield would be around 8000 kg, and this will increase with experience.

Figure 8 shows the ROI over the farm lifetime. The UK farm has a predicted 15-year cumulative net profit between −£1.50 million and £1.02 million, with an ending ROI of −42% to 61%. The increases are representative of three aspects in chronological order: (i) scaling in production in 2023; (ii) repaying the full loan amount in 2029; and (iii) upgrading to more efficient LED lighting in 2031. Despite these improvements, 50% of the scenarios fall below the required ROI threshold.

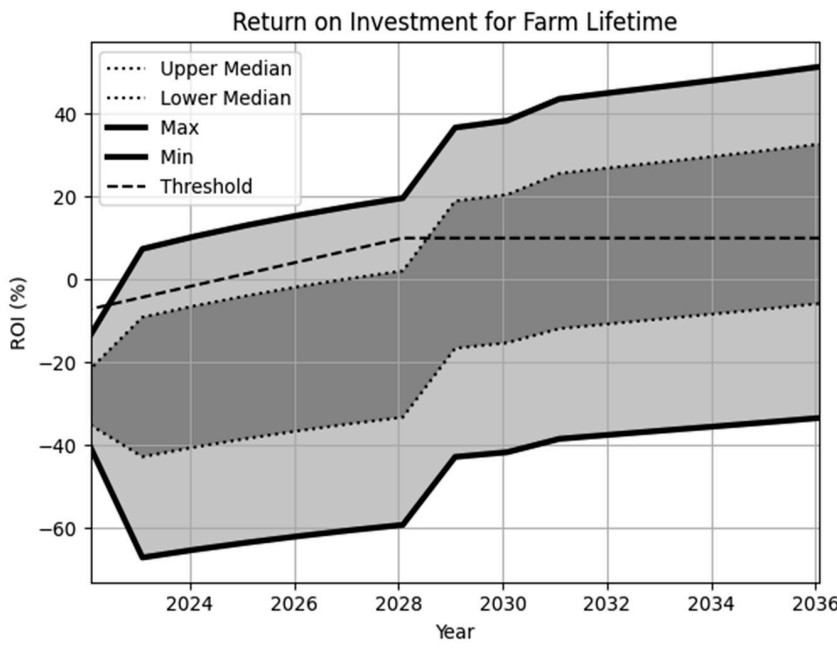

**Figure 8.** ROI potential for UK farm.

The resulting risk assessment for both the financial balance and ROI falling under their respective thresholds is shown in Figure 9. It paints an unfavourable picture of the farm, with all considered scenarios between critical and safe after a 2-year timespan indicating large levels of uncertainty and therefore no conclusion can be drawn. This prompts urgent corrective action to fix the business model, improve data collection practices and improve risk mitigation measures to reduce uncertainty. Interventions are discussed in Section 5.3.

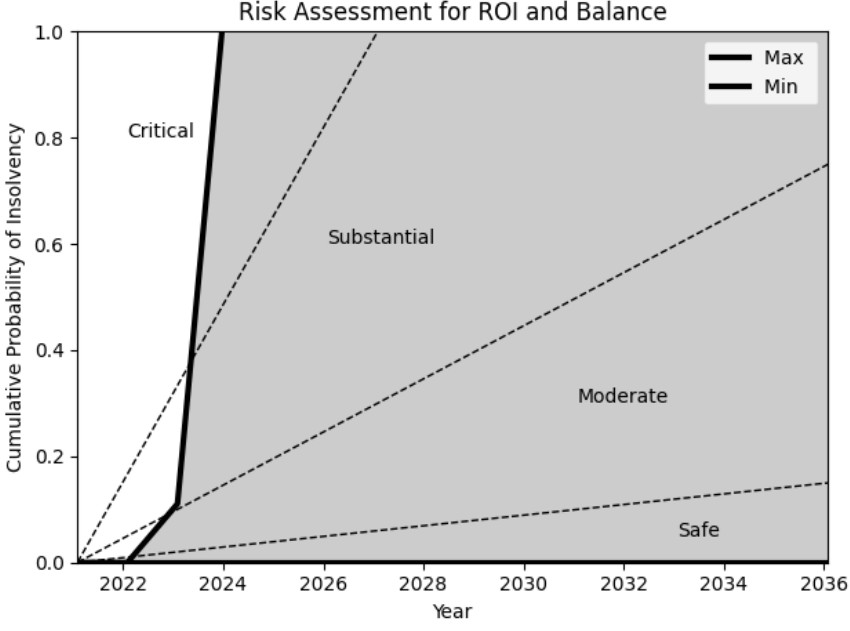

**Figure 9.** Risk profile for financial assessment for the UK farm.

### 4.2. Japanese Vertical Farm

The Japanese farm begins its operations with a financial balance of almost £570,000 and is projected over a 15-year period (see Figure 10). The graph has a narrower median compared to Figure 6 because the data provided are more precise. Over 50% of the scenarios, indicated by the dark grey area, are above the financial balance threshold, indicating a profitable business case.

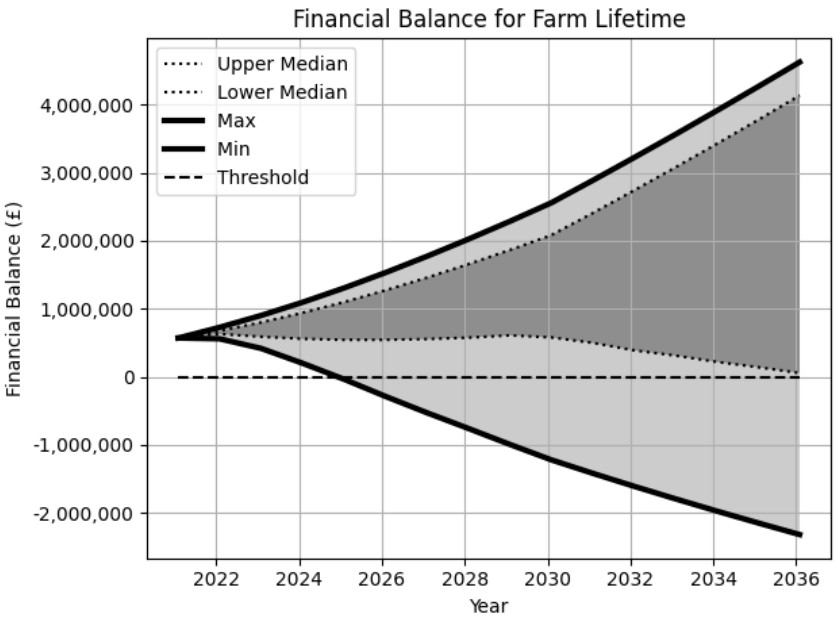

**Figure 10.** Uncertainty about financial balance for the Japanese farm over the 15-year simulation.

The annual yield for the Japanese farm for lettuce is shown in Figure 11. There is less uncertainty as the yield tracking is precise compared to the farm in Figure 7. The uncertainty remains due to improvements in crop varietals, labour efficiency and growing environment, whilst also having a risk (albeit lower than the UK farm) of pests, pathogens or customer withdrawls.

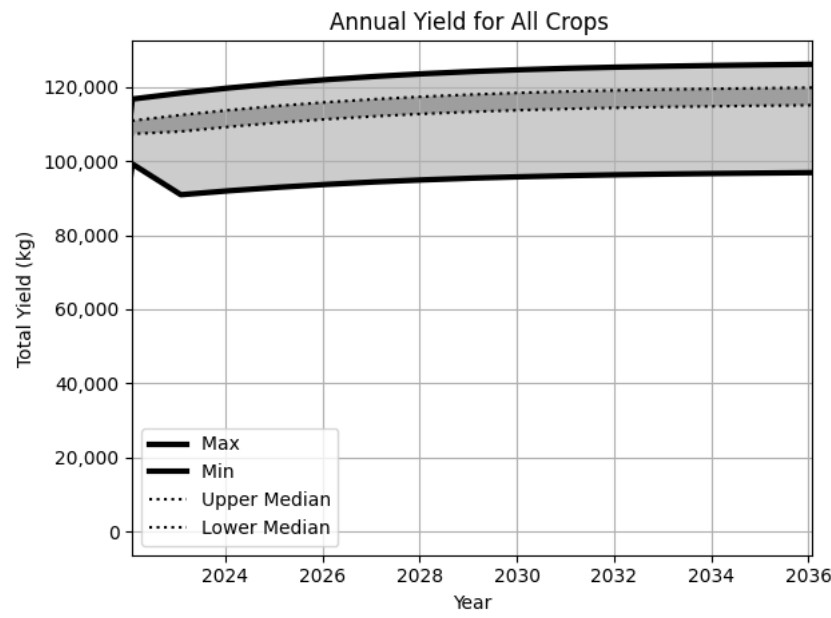

**Figure 11.** Annual yield for Japanese farm has a range between 90,000 kg and 120,000 kg. The median annual yield is 110,000 kg.

The Japanese farm has a predicted 15-year cumulative net profit between −£2.6 million and £4.6 million, with an ending ROI of 0% to 23%. Figure 12 shows the ROI over the farm lifetime. Most of the scenarios are profitable and have a positive ROI and after the light efficiency improvement in 2031, over 50% of the scenarios are above the ROI threshold.

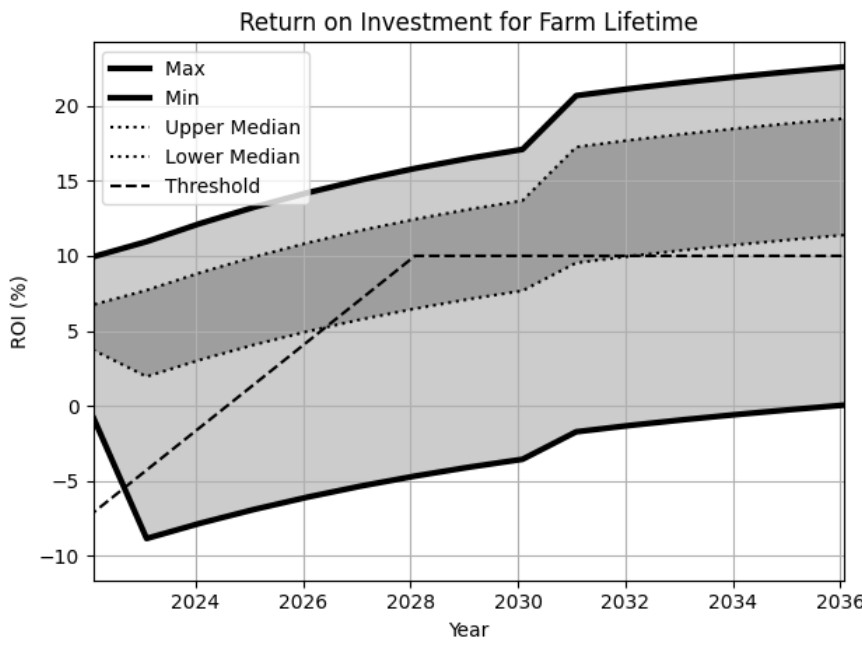

**Figure 12.** ROI potential for Japanese farm.

The resulting risk assessment for the combination of financial balance and ROI falling under their respective thresholds is shown in Figure 13. If no risks occur, the farm has 0% probability of insolvency and is in the safe region (best case). If risks such as power outages, equipment failures or crop failure (due to pests or pathogens) occur then the risk of insolvency reaches a 75% cumulative probability by 2029 (substantial risk). The future of the farm therefore lies between substantial and safe risk.

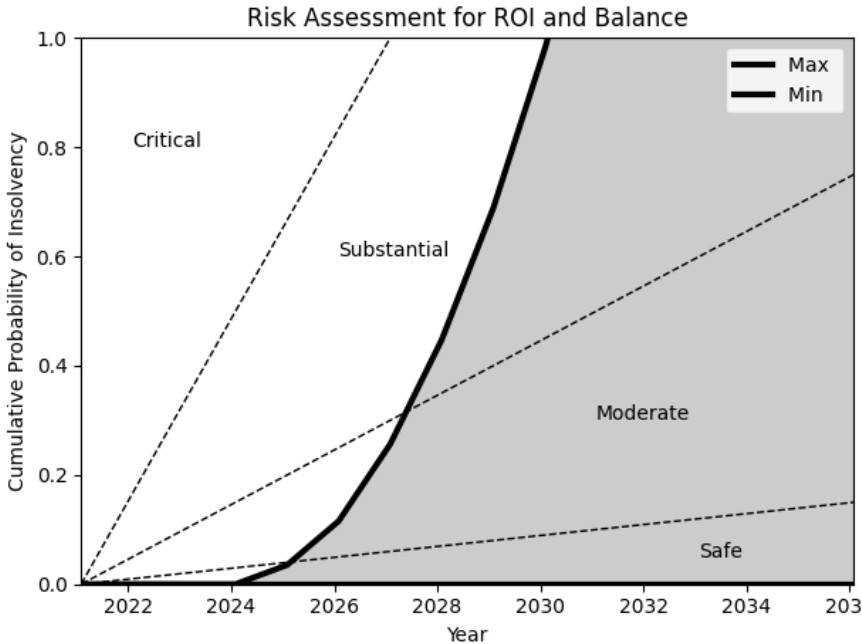

**Figure 13.** The risk profile for the cumulative probability of insolvency over 15 years shows that the Japanese farm has a safe to substantial risk profile.

## 5. Discussion

The model has simplified financial risk assessment by allowing businesses to calculate with both aleatory and epistemic uncertainty without overly precise assumptions using probability bounds. As the VF sector is still in its early stage, entrepreneurs struggle to estimate specific inputs and risks, and this method allows users to sidestep these issues. In this study, a real-life farm (UK) and a hypothetical farm (Japanese) are analysed to evaluate their risk profile in Figures 9 and 13 according to Equation (1). Default risks considered in this analysis are included in Table 2 of the method statement and analysts can create or customise their own risks using 'risk_pba.py' in the Model Library in Supplementary Materials. Users can determine whether the farm is operating at an appropriate scale and with adequate design to make a viable business model. Existing deterministic tools are not sophisticated enough to simultaneously offer best- and worst-case analysis with probability. Applying probability bounds analysis within the context of financial forecasting has never been conducted before within the academic literature. The complexity of indoor VF demands new approaches like this, as many farms have been unable to estimate economics before construction, likely resulting in either unsuccessful fundraising or wasted investments. This section discusses the two case studies, followed by proposed interventions and their effects on the UK case study. The broader implications of using this method are then described, followed by the method's limitations.

### 5.1. UK Farm

Prior expectations for the farm were made based on vertical tower vendor spreadsheets estimating 19,800 kg per year of 'leafy greens' yield extrapolated from the thesis of the vertical farming tower inventor [40,76]. Based on farm data collected for this analysis, an estimated 10,800 kg per year of lettuce will be achieved without intervention, which is 45% less than expected, resulting in drastically reduced profitability prospects. The dilemma for the UK farm is that it is currently operating at a loss and projections for both financial balance and ROI intersect below the thresholds for the majority of the lifetime of the farm. Drastic changes in the business model are required to mitigate this risk. Despite a rent-free location, low-cost labour, and subsidised energy expenditure (up to 50% off the UK average), the potential costs could still outweigh the company's revenues despite the hefty prices that they charge for produce. This indicates that subsidised bills are likely necessary components that should be sought out when developing a viable VF business model. It is worth noting that this analysis has been conducted during the coronavirus pandemic, in which many hospitality businesses are struggling. Customer focus has shifted from a business-to-business model to a business-to-consumer model, and delivering directly to homes has resulted in higher marketing, packaging and delivery costs. This may have led to a costly product and a critical risk profile. The case study was also isolated without considering other revenue streams, such as education-related income, to glean insights into the unit economics of the farm. The lack of hard data, especially for yield, has made evaluating the economics difficult for current farm activities up until now. This analysis enables computation despite unknowns and provides a quantitative evaluation to correct the course towards a financially safer risk profile.

There is a noticeable increase in positive ROI potential due to loan repayments ending and improved lighting efficiency starting in 2028 (Figure 8). However, the likelihood of ROI falling below the threshold is substantial, with over 50% of scenarios (shaded in grey) earning insufficient ROI. Further investment is required to be able to keep the farm financially afloat and make necessary changes towards economic sustainability. The model allows experimentation of potential interventions to form a roadmap to profitability. It has achieved this already during validation, as the analysis informed real business changes for the case study farm owners, such as more accurate data collection and adjustment of packaging and distribution methods.

### 5.2. Japanese Farm

Compared to the PFAL referenced [22], this analysis accounts for additional fixed costs like depreciation, staff salaries, and other costs to make it more realistic (see Table 6). Therefore, it is expected that the analysis would reveal a reduced ROI (calculated as net profit divided by capital costs) compared to the literature example. In the literature, the PFAL has a 20.5% ROI after five years, whilst this analysis predicts a −5 to 15% ROI after 5 years (50% of the farm scenarios have an ROI between 6–12.5%). The annual yield is the same as the example and is comparably higher per square-metre (117 kg per $m^2$ per year) than the UK farm (49.1 kg per $m^2$ per year). This is because the PFAL has been improved for crop varietal, crop growth recipes and labour efficiency.

The Japanese farm has a positive outlook with a risk profile between substantial (worst case) and safe (best case) in Figure 13. The unit economics are profitable, and the farm is more resilient to the risks affecting the smaller UK farm (small repairs, pest outbreaks and electrical outages). On the other hand, the Japanese farm may be more prone to labour challenges (due to a larger team size and low-cost workers), costly equipment failures and customer withdrawal (market shocks) from a supermarket for example. The average financial balance and ROI is over the threshold for the most part. However, the size of the P-box is still covering multiple zones indicating uncertainty, primarily driven by the lack of empirical data for the risks and opportunities. The risk profile is more favourable than that for the UK farm and represents an ideal farm in a more mature market. There is still a significant probability of insolvency from 2025 onwards. Changes could be made to the business model such as seeking alternative revenue streams; however, a substantial risk profile is to be expected in an innovative sector. Because the case study is hypothetical, it is not possible to say whether the risk assessment is wholly grounded in reality. Certain aspects, such as the high yield, should be probed further. If desired, the model could be used to trial other decisions and risk mitigation strategies to see how this may reduce financial risk to a safe investment.

### 5.3. Interventions to the UK Case Study

The model allows for consideration of alternative decisions to visualise how they alter the farm's business model and risk profile. The UK farm is in a situation of critical risk, and therefore interventions will be focused on this case study. The proposed adjustments could course-correct the farm (defined in Table 3) towards more favourable unit economics and a reduction in pathogen and pest risks. Moreover, diversifying revenue streams would reduce reliance on an optimised growing environment that may be difficult to achieve in a retro-fitted structure. Interventions are suggested in Table 7 based on learnings from the results in Section 4.1 and through experimentation with model inputs.

**Table 7.** Suggested interventions for UK case study.

| Intervention | Input Change | Result |
|---|---|---|
| Tailor nutrient solution composition to specific lettuce varietal | Nutrient control: medium to high | Improved yield and produce quality by ~10% [1] |
| Provide carbon dioxide enrichment | $CO_2$ enrichment: no to yes | Improved yield and produce quality by ~10% [1] |
| Improve climate control through HVAC system | Climate control: low to medium. Additional 5–20% energy costs | Improved yield by ~5% [1] and reduced likelihood of pathogens and pests [2] |
| Alter packaging solution with digital information rather than printed leaflets | Reduce cost from £1.00 to £0.70 per unit | Reduced unit costs |
| Adopt robust biosecurity protocol requiring more regular cleaning of the systems | Biosecurity control: medium to high | Reduced likelihood of pathogen outbreaks [2] |

**Table 7.** *Cont.*

| Intervention | Input Change | Result |
|---|---|---|
| Use efficient distribution channels by focusing on bulk customers | Distribution unit costs are reduced by 50% | Reduced unit costs |
| Acquire further capital funding for proposed improvements | £100,000 grant in year 2 | £20,000–30,000 additional capex |
| Utilise load shifting to optimise electricity prices (see [63]) | From £0.073–0.108 to £0.073–0.085 | Reduced unit costs |
| Introduce tours of the farm with a dedicated tour guide | £2000 revenue per month (10% increase/year) and tour guide salary budgeted | Increased revenue and mitigate risk of crop failure severely affecting income |
| Account for higher expenses associated with $CO_2$, nutrient solution, biosecurity and tour marketing | From 2% to 5% of salaries | Increased costs |

[1] See Equation (S6) in method statement, [2] see Tables S2 and S3 of method statement.

The input changes for the model in Table 7 are changed within 'main_pba_UK_Farm _interventions.py' which affect the results according to the method statement. The crop limiting factor is still not entirely understood, and crop growth factors like $CO_2$ factor and nutrient factor effects are estimated according to [28]. The effects of these adjustments can be seen in financial balance and ROI projections (Figures 14 and 15, respectively). The combination of these two metrics results in financial risk assessment shown in Figure 16.

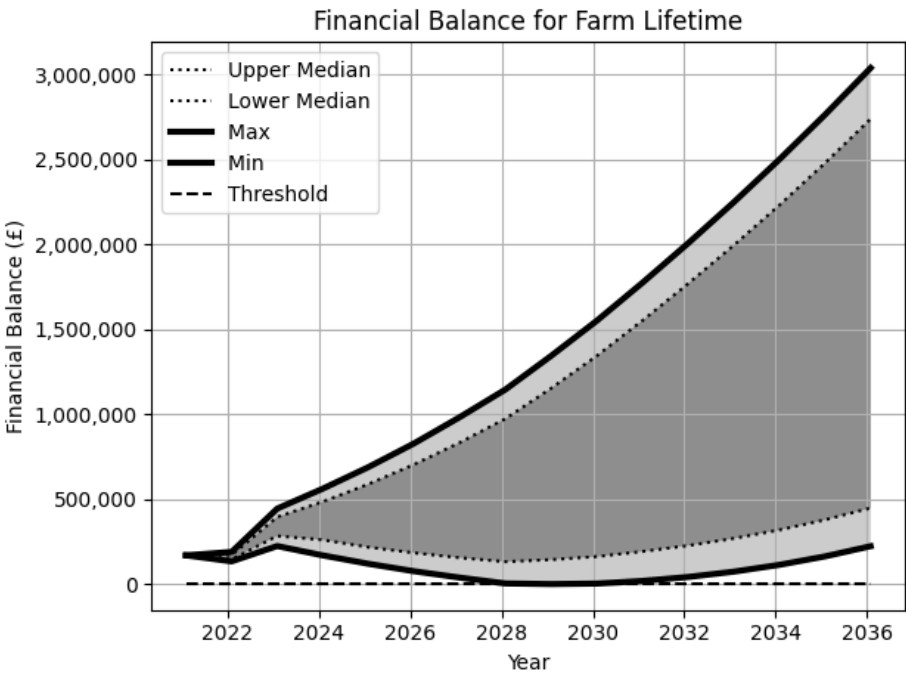

**Figure 14.** Financial balance projections for UK case study after suggested interventions.

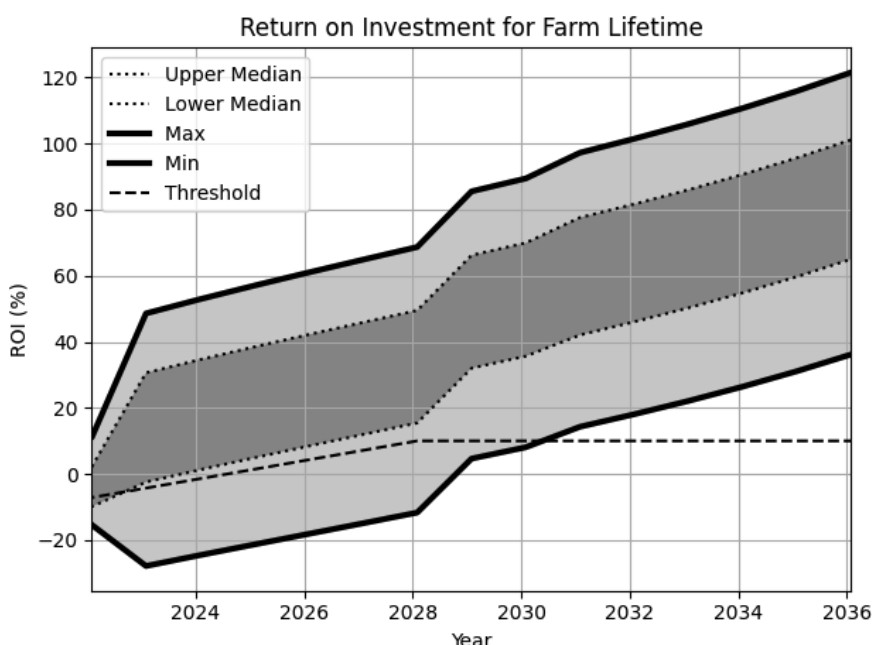

**Figure 15.** ROI projections for UK case study after suggested interventions.

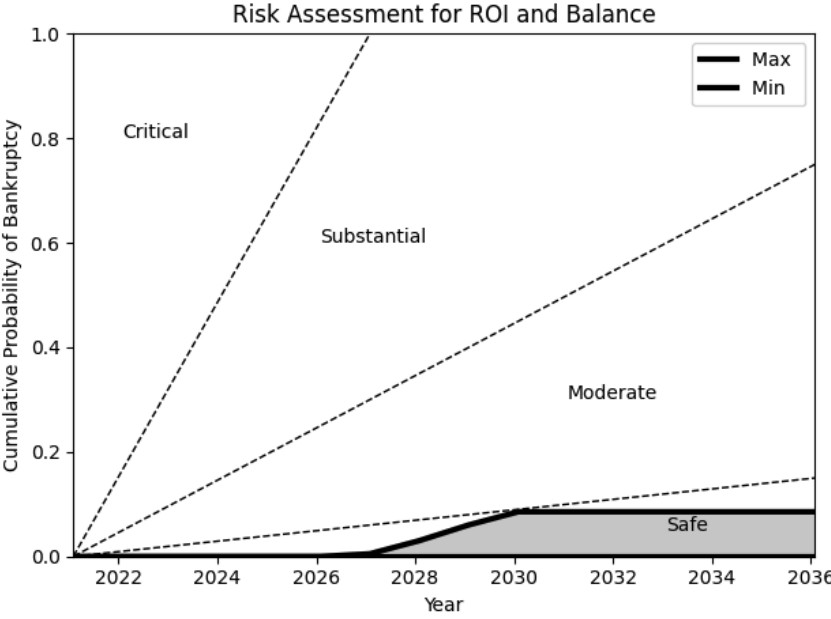

**Figure 16.** Risk profile for the probability of insolvency over 15 years shows that the UK farm is 'safe' after proposed interventions.

The post-intervention risk assessment of Figure 16 is now within the safe boundaries for both the worst- and best-case scenarios, providing a vastly more positive and certain outlook than Figure 9. There remains epistemic uncertainty that could be reduced through better tracking of yield, direct labour and consumables. This analysis is advantageous for highlighting the urgency in changing trajectory, whilst the company aims to scale up their operations. Further changes could be made, such as selecting higher-value products like speciality herbs; however, market research is required and the scenarios considered show that this is not necessary.

Another consideration is a decentralised model of distribution, whereby systems are placed at distribution points with value-added benefits for a service fee. For example, systems might be placed within a supermarket or within a restaurant and may be replenished

from the main farm facility. This is an increasingly popular farm model [77–79] and reduces distribution costs. This has been omitted from this analysis and should be integrated in future works. Other revenue streams, such as education, have been riddled with uncertainty and unpredictability due to the coronavirus pandemic but could be included. With the suggested changes in Table 7 and without considering risks, the risk profile would improve to a 0% chance of insolvency, indicating a safe investment and a highly profitable model.

### 5.4. Implications

There is a lack of hard financial data publicly available from the VF sector [18], which has led to a debate as to whether or not VF is a profitable endeavour. This model was proposed to directly address this, informing both entrepreneurs and investors to determine the viability of their plans or existing farms. The economic model is the first to enable entrepreneurs within the VF sector to evaluate their business plans whilst considering deep uncertainty. 73% of CEA founders say they would choose their equipment and crop selection differently [45] and through adequate planning this can be reduced. The iterative process of tweaking a business model becomes simplified by allowing users to assess the feasibility of their business decisions without requiring precise assumptions. It helps users understand the components necessary to construct and operate a facility, planning virtually to converge towards a viable business model. Estimating the best and worst cases with an associated probability of survival provides a transparent depiction of companies' futures. Not perfectly knowing the parameters does not preclude a quantitative analysis. Furthermore, the analysis highlights where the uncertainty lies which can help prioritise where more robust data are needed. When partial information about risks and opportunities are known, they can be accounted for selectively to plan for resilience through mitigation strategies. Using risk survey protocols, as utilised in other industries [80], could contribute to further datasets required to enhance analysis. Existing analyses described in Section 2 are unable to achieve this. For example, Monte Carlo simulations require more precise assumptions around distributions and therefore can suffer from poor accuracy.

Financial and environmental, social, and governance metrics are also provided as outputs from the model as they become increasingly sought after. Further work is required to examine other case studies across various crop types and configurations to reach conclusions on the most viable business models. This study can have global impacts by enabling entrepreneurs, investors and analysts to assess the production and economics of VF or CEA more widely without overly precise assumptions. Moreover, as probability bounds analysis captures all available information, it is possible to aggregate data of varying quality and across farm types if the uncertainty is correctly accounted for.

### 5.5. Limitations

There are a few caveats:

- The model evaluates risk assuming the condition of perfect markets (competitive prices exist for all goods in all possible contingencies). Although there exists methods to model imperfect markets [57], these have been omitted from the analysis to avoid excessive uncertainty that reduces the ability to draw any concrete conclusions.
- The model is able to compute yield without the precise user input based on Equation (S6) within the method statement. The relationship between environmental controls and yield is nuanced and this equation adapted from existing research [28] is a simplification of a crop's limiting factor [81]. As this relationship is further understood in the academic literature, this can be expanded to incorporate the limiting factor and provide a more accurate yield estimation.
- Risks and opportunities have been modelled based on anecdotal reports [14]. Meaningful distributions would require longitudinal data of adequate risk reporting (frequency and impacts). A lack of track records means that such data do not currently exist [22]. This is a primary reason for choosing probability bounds analysis, which does not require overly precise estimations. For the time being, risks and opportunities are

based on default settings; however, users are welcome to add or modify risks from their own experience and operational history.

- Two case studies have been analysed and juxtaposed to show different systems, markets, climates and scales. Further case studies are required to generate meaningful conclusions about the industry and typical risk profiles. A comparison to a state-of-the-art greenhouse with adjusted risks would give further insight into the risk profile of other production methods. However, this was out of the scope of this article.
- The model has been calibrated to compute realistic financials for both case studies [31,73]. The analysis would benefit from a more careful validation, requiring longitudinal financial data and operational histories.
- Evaluation of economies of scale would require a deeper analysis of variable costs and how they vary with production quantity across multiple farms.
- The model can compute estimated yields for various crops. However, the analysis presented only examines lettuce farms. Investigating other case studies for other crop types (micro-herbs, mushrooms, berries) may reveal different characteristics, risks and opportunities.
- Other financial indicators such as current ratio, liabilities/total assets ratio, equity/total assets ratio and cash ratio should be included in future iterations of this model.
- Currently the model predicts bankruptcy with the same method regardless of location; however, there is a dependence between explanatory variables and the country, which should be considered in future works [74].

## 6. Conclusions

Industry practitioners claim that the economic viability of vertical farms is possible with a robust business model and a focus on unit economics. However, financial viability requires demonstration and comparative financial data to have scientific validity. A significant obstacle to profitability is knowledge acquisition on how to design and run an efficient VF business. The literature calls for more robust economic analyses for vertical farms. On the other hand, there is a lack of hard data for yields, cost, risks and labour. This study handles partial information by proposing a financial risk model that incorporates the risks and uncertainty of these intricate systems to enhance accuracy.

The method described in this paper assesses economic viability and financial risk despite the lack of available production and financial data. In addition, it can be used to inform improvements in farm design towards profitable business models. The financial risk analysis and model library can be found at: https://github.com/GaiaKnowledge/VerticalFarming (accessed on 6 February 2022) as a part of a wider decision support system project [12]. It utilises probability bounds analysis combined with first-hitting-time, which has been used for other disciplines in ecology and engineering [72]. This method is applied to both real-life (UK) and hypothetical (Japanese) vertical farms.

The UK farm shows that the path to profitability requires many competing factors to be optimised. This aligns with existing research that no specific placement (urban, peri-urban, rural) with varying climate conditions results in a simple net-positive or negative result [75]. For the first time, this can be assessed with incomplete data. The results for the UK case study reveal a critical financial risk (see Figure 9) requiring drastic changes to the farm business model. Currently, the farm is operating at a loss, as the business experiments with different technologies, strategies and revenue streams. A path to profitability is being forged through trialing various interventions like further capital injection and improvements to climate control. This collectively results in a more favourable and safe risk profile. The farm operators utilised the model and the results led them to prioritise the collection of more accurate data, especially for metrics that impact profitability.

A real-life case study that shows clear profitability is required in future work to prove or disprove the claim that vertical farms can be profitable. Due to the absence of available data, a Japanese farm from the literature was also used as a hypothetical case study. The hypothetical Japanese farm offers a more resilient business model with an acceptable ROI,

but longitudinal data validation is required to determine whether the hypothetical farm is a realistic long-term scenario.

The economic sustainability of vertical farms is primarily driven by high crop yields per unit area as well as electricity, labour and depreciation costs. Despite this, it has become clear from this analysis that using an off-the-shelf system combined with benefits of free rent, low-cost electricity, low-cost labour and a premium price point, does not guarantee positive unit economics and low financial risk. The value that VF delivers to a location is significant and the aforementioned benefits should always be sought out to improve a project's profitability prospects. However, the economics should be carefully evaluated prior to construction. In reality, almost all vertical farms struggle to compare the economic feasibility of different systems and solutions but this can now be achieved more accurately with this economic risk model through allowing analysts to avoid making precise assumptions and more likely to capture true production and financial values.

This analytical research is exploratory and has been conducted on two case studies. It is challenging to draw generalised conclusions on this new industry due to the vast array of business models and proprietary systems being developed. There is no clear formula to profitability and every farm is operating within entirely different constraints (technology, market, climate, building and crop selection). This means that there is no one-size-fits-all approach to VF and each situation should be considered unique. From the model combined with available literature [18,22], it can be deduced that keys to higher profit margins can be found in: (i) scaling operations (whilst fixed costs remain the same); (ii) reducing capital costs due to maturing technology; (iii) improving labour efficiency; (iv) increasing produce quality and yield through crop genetics and growing environment optimization; (v) commanding a premium price; and (vi) reductions in costs such as subsidised rent or electrical efficiency improvement. In future works, more real-life case studies with comprehensive data of various crop types, business models and VF configurations are required to make concrete conclusions about the sector. Longitudinal data of operational histories and financial reporting would enable further validation of the model and facilitate benchmarking that can inform investment decisions. This sector has the potential to radically alter the way we grow and distribute food across the world but only if cost performance can be improved. Risk-empowering businesses, advancing technology, and sharing of data are several aspects that will accelerate this.

As industries become increasingly complex, techniques such as probability bounds analysis already used in other disciplines will be helpful in financial modelling. There is no dispute that the financial futures of start-up businesses are uncertain. Forecasting deterministically or through Monte Carlo simulations provide a simplistic and sometimes inaccurate view. What happens when data about precise model distributions or exact parameters are not available? This is the case for vertical farming. A method such as probability bounds analysis facilitates these computations to open up a new realm of scenario analysis and financial risk management. Vertical farming is only one complex industry of many that could benefit from such a method.

This is the first academic study applying financial risk assessment to vertical farming. By building the foundation of literature on risk in vertical farming, investors can begin to understand this emerging market which will increase access to favourable types of capital. This work enables entrepreneurs, investors, and analysts to assess the production and economics of VF or CEA more widely without overly precise assumptions. Moreover, as probability bounds analysis captures all available information, it is possible to aggregate data of varying quality and across farm types if the uncertainty is correctly accounted for.

**Supplementary Materials:** The following supporting information can be downloaded at: https://www.mdpi.com/article/10.3390/su14095676/s1, Supplementary Data, Method Statement [82–88], Model Library.

**Author Contributions:** Conceptualisation, F.J.B.d.O. and S.F.; methodology, F.J.B.d.O.; software, F.J.B.d.O., S.F., J.M.H.T. and N.G.G.; validation, F.J.B.d.O., J.M.H.T. and P.D.M.; formal analysis,

F.J.B.d.O.; investigation, F.J.B.d.O.; resources, F.J.B.d.O.; data curation, F.J.B.d.O.; writing—original draft preparation, F.J.B.d.O.; writing—review and editing, F.J.B.d.O., R.A.D.D. and S.F.; visualisation, F.J.B.d.O.; supervision, S.F. and R.A.D.D.; project administration, F.J.B.d.O.; funding acquisition, P.D.M. and J.M.H.T. All authors have read and agreed to the published version of the manuscript.

**Funding:** This research was partially funded by the Low Carbon Eco-Innovatory and European Regional Development Fund, grant number 22R16P00045.

**Institutional Review Board Statement:** Not applicable.

**Informed Consent Statement:** Not applicable.

**Data Availability Statement:** The supporting data are openly available alongside reported results. These can be found in two places, Supplementary Materials (supplementary_data.pdf) and the open-source repository found online at: https://github.com/GaiaKnowledge/VerticalFarming (accessed on 6 February 2022). The UK case study inputs are found as 'Current_Financial_Model_FU_v1.xlsx', processed in 'main_pba_UK_Farm.py' alongside results 'results_UK.xlsx'. The Japanese case study inputs are found as 'Current_Financial_Model_JP_PFAL.xlsx', processed in 'main_pba_JP_PFAL.py' alongside results 'reuslts_JPFA.xlsx'. The UK farm post interventions is processed as 'main_pba_UK _Farm_interventions.py' alongside results 'results_UK_post.xlsx'. Default data on risks is found at 'risk_pba.py'.

**Acknowledgments:** We would like to acknowledge the team at Farm Urban for assisting in collection of the data for the UK case study and friends and family that indirectly helped this research come to fruition over several challenging years during the coronavirus pandemic.

**Conflicts of Interest:** The authors declare one conflict of interest. Two co-authors are the co-founders of a vertical farming research company, Farm Urban. Their input enabled the inclusion of the only real financial case study within the literature. Their involvement was limited to helping to collect and validate the data required and produced by the model. They had no role in the design, analyses or interpretation of the data.

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
