# Peer review of "How High Is High Enough? Assessing Financial Risk for Vertical Farms Using Imprecise Probability"

_sustainability, doi:10.3390/su14095676_

Round 1

Reviewer 1 Report

The paper is rewriten according to standards. Now it contains all part as should be in scientific paper. Methodology and results are sufficiently described followed by discussion part. I only recommend to add more up to date literature: 

Kovacova, M., and Lăzăroiu, G. (2021). “Sustainable Organizational Performance, Cyber-Physical Production Networks, and Deep Learning-assisted Smart Process Planning in Industry 4.0-based Manufacturing Systems,” Economics, Management, and Financial Markets 16(3): 41–54. doi: 10.22381/emfm16320212.

Olah, J., Tiron Tudor, A., Pashkus, V., Alpatov, G. (2021). Preferences of Central European consumers in circular economy, Ekonomicko-manazerske spektrum, 15(2), 99-110.

Kovacova, M., and Lewis, E. (2021). “Smart Factory Performance, Cognitive Automation, and Industrial Big Data Analytics in Sustainable Manufacturing Internet of Things,” Journal of Self-Governance and Management Economics 9(3): 9–21. doi: 10.22381/jsme9320211.

Tijani, A.A., Osagie, R.O., Afolabi, B.K. (2021). Effect of strategic alliance and partnership on the survival of MSMEs post COVID-19 pandemic, Ekonomicko-manazerske spektrum, 15(2), 126-137.

Valaskova, K., Ward, P., & Svabova, L. (2021). Deep Learning-assisted Smart Process Planning, Cognitive Automation, and Industrial Big Data Analytics in Sustainable Cyber-Physical Production Systems, Journal of Self-Governance and Management Economics. 9(2), 9–20.

Durana, P., Perkins, N., & Valaskova, K. (2021). Artificial Intelligence Data-driven Internet of Things Systems, Real-Time Advanced Analytics, and Cyber-Physical Production Networks in Sustainable Smart Manufacturing. Economics, Management, and Financial Markets, 16(1), 20–30.

Reviewer 2 Report

The manuscript has conducted economic viability and risk assessment of vertical farming in the absence of available production, risk, and financial data. Besides, the manuscript also aimed to find our risk profile and use risk assessment tools to inform a profitable business model. This is indeed an interesting paper involving rigorous analysis acknowledging its shortcoming as well.

I believe some sort of balanced should be maintained between the explicitly stated three objectives. Largely, there is a lack of analysis to address the second objective. 

Some more explanation of the graphs is warranted.

I also found many self-citations that may not be appropriate.

Please refer to the attachment for some specific comments.

Best wishes

Reviewer 3 Report

This manuscript has the potential to become an interesting paper, but more work is required.

My initial recommendation for rejection remains for the same reasons: a poor understanding of the problem, lack of review of relevant peer-reviewed literature, no originality in methodology and any additional information offered is based on unpublished research. Finally, results are not addressing the issues described in the introduction, are not relevant to any other farm or the overall literature and thus do not contribute to the body of knowledge in this area.  

The comparative results from these two unique farms does not offer a general advice to the industry or to researchers. It rather only applies a previously published model to two very different scenarios, one heavily subsidized school-based farm and the Japanese Plant Factory, a high-tech existing R&D facility (not hypothetical), neither being a representative of an average farm in this industry.

Although the conclusion opens with the argument that “entrepreneurs struggle to estimate specific inputs and risks”, results are presented in the general form of ROI and profits, which does not address the difficulty in estimating risks for specific inputs. The simplification of results down to final ROI or profits makes results mute since these are not comparable farms between themselves or to the overall industry.

I recommend thorough re-writing of the paper, starting with an introduction that uses relevant, up to date and peer-reviewed literature, and a restructuring of the analysis. Based on accurate perception of the problem, the analysis would be interesting to readers if it provided analysis of specific risks.

More specific comments:

The paper builds reasoning upon assumptions which are either outdate or on news articles based on anecdotal perceptions of a few individuals (for example, their own publication which is heavily based on arguments from Liotta et al 2017). As a consequence, the description of the industry and justification for the work is based on a misunderstood view of the industry and relevant literature. An example of misunderstanding of the industry is the claim on lines 118-124 that software systems are few and flawed. On line 105 “The sector is notorious for being closed” this is a common criticism to the U.S. industry. Lines 173-174: “no analyses consider automated systems”. The argument contradicts previous citations of Kozai’s work and VF 2.0. Automation is a general word, and it can take many levels. The industry has grown rapidly and is no longer considered at “early stages” but rather at its teenage years.

The criticism of comparative analysis between GH and VF (127-134) is groundless, since greenhouses also make use of supplemental lighting. The study being criticized is clearly describing an example of a GH and a VF that use the same type of equipment overall, with comparable HVAC and LED efficiency, which allows for a ceteris paribus comparison between productivity and yield.

Why is the likelihood of ROI falling below threshold if loans have been repaid and lighting system was improved? What was the improvement (lines 473-475)?

It was stated (line 482) that the Japanese farm is resilient to “most risks”. Which ones and which are their weaknesses? Why would alternative revenue streams benefit the farm (line 489) – which risk does that refer to?

Are suggestions from Table 7 based on the risk assessment? If so, they need to be described in relation to the base data used in the model. For example, “tailor nutrient solution to lettuce” is a very general advice. It reads as increasing or choosing appropriate nutrient. Then again, the suggested input change relates to nutrient control. Is that a suggestion for automation? Which input was used to lead to that suggestion? In fact, the whole discussion brought up by table 7 is incomplete. The paper needs a new section explaining the impact of those changes into specific risks. The suggestion for quality increase, improved yield, reduced pathogens, reduce unit costs and increased revenue, at a price of increasing capital investment and costs seem a straightforward advice to any company. How do they relate with the model and how do they affect the weight on each risk on the overall performance of the farm? That is information that could be applied to other scenarios.

Finally, the second version of the manuscript shows changes in the reference list, but citations were not updated accordingly. For example, the discussion following table 1 is inconsistent with literature cited.

Reviewer 4 Report

The paper focuses on a very actual but less researched topic. Risk issues in the case of vertical farms are essential and their nature differs from traditional agriculture,

The title is correct and reflects the content of the paper. The abstract is well written, comprehensive, and compact, contains all essential info on the paper.

The introduction is acceptable, indicates the reasons and goals of the research and the context as well.

The literature review is a great piece, that overviews the latest international sources of the field and subfields.

The methodology is demonstrated and introduced in very detail and supports the results. The methodological toolset is selected appropriately.

The results are clear and well understandable. provides a good base for conclusions and implications.

I recommend to publish this paper as it is, without any changes.

Reviewer 5 Report

This paper titled ‘How High is High Enough? Assessing Financial Risk for Vertical Farms Using Imprecise Probability’ can be seen as a contribution to research and has good publishing potential because the research topic is interesting and applicable to policy. But unfortunately, still, it is not in the condition that it could be accepted for publication after revisions. The authors are suggested to improve it and submit it again.

  • Your problem statement should be clear. Although rationale for the study is given but it needs to be expanded. Clearly mention the need for project (how this study will be a new contribution in the field of financial risk for vertical Farms). Further, write research hypotheses after defining the research gap.
  • Summarize the literature review in a table, with columns specifying: authors, source, year, the model used, objective, and results. In the end, compare your results with the ones obtained by other, if such studies exist. You can incorporate stated items in your Table 1 by adding columns.
  • It is important to emphasize, what type of model selection test you have conducted. Moreover, explain why other models were not used.
  • It is recommended that the result of this study should be compared to the results obtained by other authors in published articles on a similar topic and country recently.
  • Include a section on consistencies of the results with prior expectations.
  • Support the claim throughout the article with published sources.
  • Include a section on how the results of this study can have global impacts other than Vietnam.

Round 2

Reviewer 3 Report

This manuscript shows a poor understanding of the problem, lack of review of relevant peer-reviewed literature, and any additional information offered is based on unpublished research. Finally, results are not addressing the issues described in the introduction, are not relevant or replicable to any other farm in the industry or the overall literature and thus do not contribute to the body of knowledge in this area.  

The comparative results from these two unique farms does not offer a general advice to the industry or to researchers. It rather only applies a previously published model to two very different scenarios, one heavily subsidized school-based farm and the Japanese Plant Factory, a high-tech existing R&D facility (not hypothetical), neither being a representative of an average farm in this industry.

Although the conclusion opens with the argument that “entrepreneurs struggle to estimate specific inputs and risks”, results are presented in the general form of ROI and profits, which does not address the difficulty in estimating risks for specific inputs. The simplification of results down to final ROI or profits makes results mute since these are not comparable farms between themselves or to the overall industry.

I recommend thorough re-writing of the manuscript, starting with an introduction that uses relevant, up to date and peer-reviewed literature, and a restructuring of the analysis. Based on accurate perception of the problem, the analysis would be interesting to readers if it provided analysis of specific risks.

More specific comments:

The paper builds reasoning upon assumptions which are either outdate or on news articles based on anecdotal perceptions of a few individuals (for example, their own publication which is heavily based on personal arguments presented in a discussion panel - Liotta et al 2017). As a consequence, the description of the industry and justification for the work is based on a misunderstood view of the industry and relevant literature. An example of misunderstanding of the industry is the claim on lines 118-124 that software systems are few and flawed. On line 105 “The sector is notorious for being closed” this is a common criticism to the U.S. industry. Lines 173-174: “no analyses consider automated systems”. The argument contradicts previous citations of Kozai’s work and VF 2.0. Automation is a general word, and it can take many levels. The industry has grown rapidly and is no longer considered at “early stages” but rather at its teenage years.

The criticism of comparative analysis between GH and VF (127-134) is groundless, since greenhouses also make use of supplemental lighting. The study being criticized is clearly describing an example of a GH and a VF that use the same type of equipment overall, with comparable HVAC and LED efficiency, which allows for a ceteris paribus comparison between productivity and yield.

Why is the likelihood of ROI falling below threshold if loans have been repaid and lighting system was improved? What was the improvement (lines 473-475)?

It was stated (line 482) that the Japanese farm is resilient to “most risks”. Which ones and which are their weaknesses? Why would alternative revenue streams benefit the farm (line 489) – which risk does that refer to?

Are suggestions from Table 7 based on the risk assessment? If so, they need to be described in relation to the base data used in the model. For example, “tailor nutrient solution to lettuce” is a very general advice. It reads as increasing or choosing appropriate nutrient. Then again, the suggested input change relates to nutrient control. Is that a suggestion for automation? Which input was used to lead to that suggestion? In fact, the whole discussion brought up by table 7 is incomplete. The paper needs a new section explaining the impact of those changes into specific risks. The suggestion for quality increase, improved yield, reduced pathogens, reduce unit costs and increased revenue, at a price of increasing capital investment and costs seem a straightforward advice to any company. How do they relate with the model and how do they affect the weight on each risk on the overall performance of the farm? That is information that could be applied to other scenarios.

Finally, the second version of the manuscript shows changes in the reference list, but citations were not updated accordingly. For example, the discussion following table 1 is inconsistent with literature cited.

Author Response

--

This manuscript is a resubmission of an earlier submission. The following is a list of the peer review reports and author responses from that submission.

Round 1

Reviewer 1 Report

I do not recommend this manuscript for publication as further work is required to produce a novel analysis that contributes to the body of knowledge in this area of research and is relevant to the industry. As it is, this manuscript offers a comparative analysis of financial risks between two very distinct types of farms and builds assumptions on either outdate data or on news articles based on anecdotal perceptions (and sometimes controversial) of a few individuals. The manuscript also relies heavily on unpublished research.
Specifically, articles aggregated for the introduction, section 2 "related works", and later to define assumptions, refer to different crops (rice and lettuce, for example) and distinct production systems, such as container farms, greenhouses, and vertical farms, all of which using different types and levels of technology and automation. Although similar in some respects, these systems’ performances are ultimately not directly comparable but rather only in specific aspects.
In 2.2 "Risks and Opportunities", the reasons for ceasing trading are not exclusive to vertical farming, some are specific to container farms, which have their particular issues, and some seem to have been misunderstood by the authors. Overall, I recommend re-writing all tables with better descriptions and descriptions that also match the flow of the analysis. For example, "poor quality capital" should probably be redefined if the purpose was to describe sources of investment in vertical farms and their expectations. Define better "lack of adequate knowledge", knowledge about what? Is this referring to access to trained workforce? If so, how does that relate with "underestimated labor costs" or with "loss of expertise" (Table 2)?  Also, I recommend further research for this section, as there are more pressing issues affecting this industry which are not listed. 
Further research is required throughout the analysis but also needed to complete the section on “suggested interventions” in order to make these results meaningful. 

Reviewer 2 Report

The manuscript has assessed the economic viability and financial risk of vertical farming despite the lack of available production and financial data. This is indeed an interesting paper involving rigorous analysis acknowledging its shortcoming as well.

I found this paper nicely written, with very few typos. 

I have one minor query and one suggestion. 

Is it possible to have the median annual yield of 11,000kg when the range is between 90,000kg and 120,000kg (lines 411-412)?

I suggest authors could point out potential interventions in the abstract, such as accurate data collection, adjustment of packaging, and distribution methods (Lines 464-465) or "further capital injection and improvements to climate control" Lines (597-598) or the most critical one listed in Table 7 in the case of the UK farm; and changes in the business model seeking alternative revenue streams in case of Japan (?)

Discussing Table 7 taking reference to Table 3 will help to relate the suggested interventions in relation to the existing state of the technology.

Best wishes

Reviewer 3 Report

The paper is properly written based on the relevant and up-to date literature review.

I only recommend to add some scietific papers to the literature: 

Kovacova, M., and Lewis, E. (2021). “Smart Factory Performance, Cognitive Automation, and Industrial Big Data Analytics in Sustainable Manufacturing Internet of Things,” Journal of Self-Governance and Management Economics 9(3): 9–21. doi: 10.22381/jsme9320211.

Milanov, A. (2020). Risk measurement and evaluation in RFI and RFP processes at the Bulgarian mobile telecommunication operators, Ekonomicko-manazerske spektrum, 14(2), 24-35.

Osagie, R.O. (2020). Financial Inclusion: A panacea for attaining sustainable development in developing countries like Nigeria, Ekonomicko-manazerske spektrum, 14(2), 1-11.

Kovacova, M., Kliestik, T., Valaskova, K., Durana, P., & Juhaszova, Z. (2019). Systematic review of variables applied in bankruptcy prediction models of Visegrad group countries. Oeconomia Copernicana, 10(4), 743-772.

Methodology used in the paper is suitable, the sources of information and the measurement tools are adequately defined.

Gained results are described followed by the discussion part. I would like to highlight limitations of the study presented. The conclusion part also contains possible future research.